# Sports-Related Concussion Assessment: A New Physiological, Biomechanical, and Cognitive Methodology Incorporating a Randomized Controlled Trial Study Protocol

**DOI:** 10.3390/biology12081089

**Published:** 2023-08-04

**Authors:** Gareth Irwin, Matthew J. Rogatzki, Huw D. Wiltshire, Genevieve K. R. Williams, Yaodong Gu, Garrett I. Ash, Dan Tao, Julien S. Baker

**Affiliations:** 1Faculty of Sports Science, Ningbo University, Ningbo 315010, China; girwin@cardiffmet.ac.uk (G.I.); guyaodong@hotmail.com (Y.G.); jsbaker@hkbu.edu.hk (J.S.B.); 2Research Academy of Medicine Combining Sports, Ningbo No.2 Hospital, Ningbo 315010, China; 3Cardiff School of Sport and Health Sciences, Cardiff Metropolitan University, Cardiff CF52YB, UK; hwiltshire@cardiffmet.ac.uk; 4Sport and Health Interdisciplinary Group in Movement & Performance from Acute & Chronic Head Trauma (IMPACT) Group, Cardiff Metropolitan University, Cardiff CF52YB, UK; rogatzkimj@appstate.edu (M.J.R.); g.k.r.williams@exeter.ac.uk (G.K.R.W.); garrett.ash@yale.edu (G.I.A.); 5Department of Health & Exercise Science, Appalachian State University, Boone, NC 28608, USA; 6Department of Sport and Health Sciences, University of Exeter, Exeter EX44QJ, UK; 7Section of General Internal Medicine, Yale School of Medicine, Yale University, New Haven, CT 06510, USA; 8Center for Pain, Research, Informatics, Medical Comorbidities and Education Center (PRIME), VA Connecticut Healthcare System, West Haven, CT 06510, USA; 9Department of Government and International Studies, Hong Kong Baptist University, Hong Kong 999077, China; 10Centre for Health and Exercise Science Research, Hong Kong Baptist University, Hong Kong 999077, China

**Keywords:** blood-based biomarkers, motor control, cognitive function, skill, biomechanics

## Abstract

**Simple Summary:**

Following injury, players may seem medically fit to return to play, but skill levels may have deteriorated due to cognitive decline, resulting in the deterioration of cognitive abilities and skills. This suggestion may explain in part the reoccurrence of head injury linking concussive events to future concussive events. As such, this study’s aim will be to investigate performance and physiological changes in rugby players post-game for head trauma and injury.

**Abstract:**

Background: Taking part in moderate-to-vigorous exercise in contact sports on a regular basis may be linked to an increase in cerebrovascular injury and head trauma. Validated objective measures are lacking in the initial post-event diagnosis of head injury. The exercise style, duration, and intensity may also confound diagnostic indicators. As a result, we propose that the new Interdisciplinary Group in Movement & Performance from Acute & Chronic Head Trauma (IMPACT) analyze a variety of functional (biomechanical and motor control) tests as well as related biochemistry to see how they are affected by contact in sports and head injury. The study’s goal will be to look into the performance and physiological changes in rugby players after a game for head trauma and injury. Methods: This one-of-a-kind study will use a randomized controlled trial (RCT) utilizing a sport participation group and a non-participation control group. Forty male rugby 7 s players will be recruited for the study and allocated randomly to the experimental groups. The intervention group will participate in three straight rugby matches during a local 7 s rugby event. At the pre-match baseline, demographic and anthropometric data will be collected. This will be followed by the pre-match baseline collection of biochemical, biomechanical, and cognitive-motor task data. After three consecutive matches, the same measures will be taken. During each match, a notational analysis will be undertaken to obtain contact information. All measurements will be taken again 24, 48, and 72 h after the third match. Discussion: When the number of games increases owing to weariness and/or stressful circumstances, we expect a decline in body movement, coordination, and cognitive-motor tasks. Changes in blood biochemistry are expected to correspond to changes in biomechanics and cognitive-motor processes. This research proposal will generate considerable, ecologically valid data on the occurrence of head trauma events under game conditions, as well as the influence of these events on the biological systems of the performers. This will lead to a greater understanding of how sports participants react to exercise-induced injuries. This study’s scope will have far-reaching ramifications for doctors, coaches, managers, scientists, and sports regulatory bodies concerned with the health and well-being of athletic populations at all levels of competition, including all genders and ages.

## 1. Introduction

Sports-related concussion (SRC) diagnosis is encumbered by a lack of validated, objective diagnostic tests [1]. Currently, the Sport Concussion Assessment Tool-5th edition (SCAT5) [2] is the standard instrument used to diagnose SRC at the side-line and pitch side of athletic and sporting events. However, research has suggested that exercise is a confounding variable not accounted for when using the SCAT5 [3,4]. This is concerning because the baseline SCAT5 is typically administered under resting conditions but is used to diagnose SRC following physical exertion during and after sports activity. In addition to the SCAT5, biochemical biomarkers have been suggested as objective markers to aid in SRC diagnosis. Similar to the SCAT5, potential SRC diagnostic biomarkers may also be confounded by exercise. Therefore, a particularly important goal of this study is to examine several biomechanical tests, motor control tests, and biomarkers to investigate whether they are influenced by contact in sports and the prevalence of injuries to the head. This will be achieved by investigating movement and coordination tests, cognitive tests, visual tests, and a variety of blood-based biomarkers prior to and after rugby match performance, with the specific aim of investigating performance degradation.

The blood-based biomarkers included in the analysis and evaluation will include protein S100B (S100B), creatine kinase (CK), cardiac troponin T (CTT), cortisol (C), neuron-specific enolase (NSE), brain-derived neurotrophic factor (BDNF), glial fibrillary acidic protein (GFAP), and ubiquitin carboxyl-terminal esterase L1 (UCHL1). Protein S100B has been shown to increase after a concussion [5,6,7], following punches and kicks to the head in boxing [8] and karate [9], after playing a contact sport [5,7,10,11,12,13,14,15,16,17,18,19,20,21], and after exercise [22,23,24,25,26,27,28]. Past research has shown that S100B can be used as a marker of brain injury since this protein is housed in the astrocytes; however, its ability to diagnose SRC appears to be confounded by exercise since S100B is also located in skeletal muscle [29]. Creatine kinase will be measured as an indicator of skeletal muscle damage [30] to aid in distinguishing S100B release caused by skeletal muscle injury.

Although cardiac troponin is typically used as an indicator of myocardial injury [31], it has also been shown to predict mortality rates from brain trauma in clinical environments [32,33,34]. Cardiac troponin will be investigated as a potential cardiovascular damage marker in rugby athletes. We expect minimal nonclinical elevations in cardiac troponin levels post-exercise as a result of exercise intensity and not cardiac pathology [35]. Cortisol is a biomarker related to stress that increases as a result of head injury [36] and also in response to head contact during boxing [8]. However, cortisol has not been shown to increase in contact sports in the absence of SRC [10,37]. Whether cortisol increases because of rugby play in the absence of SRC will be a partial focus of investigation in this study.

Neuron-specific enolase provides a diagnostic indication of brain trauma and has been shown to increase post-match in boxers [8,38], karate practitioners [9], soccer players [18,19], and American football players [10]. However, the use of NSE as a brain trauma biomarker has been questioned due to the absence of significant NSE increases in post-game hockey [20], as well as in concussed hockey players [12]. Measuring NSE pre- and post-rugby matches will help distinguish whether NSE consistently increases due to playing a contact sport in the absence of SRC. Furthermore, it is necessary to consider the implications of increased exercise on cognitive function and brain health.

Exercise has been shown to improve memory and cognitive function [39]. These improvements are possibly related to the increase in BDNF caused by exercise training. BDNF is important in neuronal growth and maintaining synaptic connectivity in the brain [40]. Since BDNF plays a critical role in brain health, it may be beneficial for BDNF to be up-regulated following brain injury to enhance recovery. However, there is a paucity of research on BDNF concentrations after brain injury. To date, there have been three studies investigating BDNF in brain injury, and all were performed on children [41,42,43]. One indicated that BDNF increased after a concussion in children [41], with the other two studies observing mixed results, with BDNF levels increasing in correlation with the severity of the brain injury [42,43]. A single bout of exercise has been shown to increase serum BDNF levels in sedentary subjects [44,45], recreationally active subjects [46,47], and trained cyclists [48,49], but increases in serum BDNF after a contact sport in trained athletes have not yet been investigated. GFAP and UCHL1 will be investigated, as these two proteins have been shown to distinguish between computed tomography (CT)-positive and CT-negative brain injury [50] and have been approved for use in the assessment of mild traumatic brain injury (mTBI) [51]. There is clearly a need to compose a test batch for the assessment of brain health.

In order to consider how to determine the most sensitive and specific test for the symptoms of SRC, we propose that an interdisciplinary approach would be powerful in determining the levels of changes in the biological system, where the dynamics of change are consistently present. Changes in cognitive, visual, and movement functions are symptoms of SRC, and it has been suggested that the multimodal assessment of these has better sensitivity than balance or cognitive tests alone [52]. For example, based on elements of the SCAT5, short-term memory, balance, and symptom severity are rated. However, while the test-retest reliability of the SCAT5 as a whole is reported to be moderate to high, the reliability coefficients of SAC and mBESS (rs = 0.58 and 0.40, respectively) are low [53]. Partly, this is suggested to be due to individual variability and the interval-scale scores of the tests. However, currently, there are no quantitative tests for cognitive, visual, or movement function included in the SCAT5, which limits the generation of precise numerical values that can be used to identify and characterize symptoms of SRC.

Executive function has been examined in line with neurological conditions [54] and is relevant to the study of acute and chronic effects of head trauma [55,56,57]. Executive function and its relationship with decision making based on specific environmental cues that underpin motor behaviors are key in sports performance and injury risk. Evidence linking concussive events to future concussive events [58] and twice the likelihood of obtaining another musculoskeletal injury [59] suggests that executive function and aspects of neuromuscular and movement functions are impaired after concussive incidents. However, there is a need to study the acute and chronic effects of mTBI/SRC individually, as well as holistic measures of body movement and coordination, to facilitate the identification of the most sensitive and specific indicators of injury and brain function.

Gross motor function tests are currently based on standing posture and gait, and these are tasks fundamental to daily activity. While technology-based data capture for balance control, such as the clinical measure of the mBESS, is being increasingly published, Buckley et al. (2018) highlight reliability and sensitivity issues [60]. These may be overcome with the use of sensor-based balance measures, particularly in the mediolateral direction [61]. Research exploring the effect of a concussion on walking gait presents conflicting conclusions. Manaseer et al.’s (2020) systematic review suggested that individuals who have had a concussion may wobble more in the frontal plane and walk slower than healthy controls [62]. Fino et al. (2018) agreed that the literature implies that gait is abnormal shortly after a concussion or mTBI, and that dual-task and complicated gait tests have special potential for detecting impaired locomotor function [63,64]. However, there is little agreement on the most important gait metrics for SRC diagnosis and therapeutic intervention. In addition, since locomotion is a fundamental motor skill that does not stress the motor control system of a healthy individual, even with dual-task paradigms, it might not be sensitive or specific enough to show changes in motor control function due to SRC in highly trained athletes. Therefore, we propose that a complex sports-specific skill that includes an element of decision making might be more effective for detecting changes in motor control function after SRC in sports players. The inclusion of an ecologically valid throwing task captured using automated motion capture with synchronized external force plates will facilitate the subsequent quantification of joint and segmental kinematics and kinetics and will allow insights into changes in coordination patterns and joint moments and power. This study will provide the first evidence of the effect of head injury on the technique of performing a sports-specific gross motor skill.

In order to understand the facets of motor control function that may underpin changes in the performance of gross motor skills, it is important to perform a number of tests of executive function and specific aspects of perception-action coupling. For example, reaction time is a promising indicator of visuo-motor deficits following SRC [65], which is quick and easy to quantify. Finger-tapping and fine motor skill tasks (f/e the Perdue Pegboard test) [66,67,68,69] have been used widely in cognitive psychology to identify neurological conditions. In order to understand the facets of motor control function that may underpin changes in the performance of gross motor skills, we will perform a number of perception-action and physical tasks. Specifically, we will explore reaction time, executive function (Stroop word test), and fine motor skills (finger tapping and Perdue Pegboard).

The study’s aim, therefore, is (1) to examine changes in biomarker concentrations and biomechanical and cognitive-motor abilities of elite rugby players across three-match tournaments until 72 h after the final game and (2) to explore the relationship between subconcussive head impacts and their intensity and changes in blood biomarker concentrations, biomechanical tasks, and cognitive-motor tasks.

It is hypothesized that (1) the whole-body movement, coordination, and cognitive-motor tasks will decrease significantly with the duration of games played because of fatigue and/or concussive events; (2) blood biomarkers (i.e., CK, cardiac troponin T, NSE, S100B, BDNF, GFAP, and UCHL1) will increase significantly after rugby tournaments; (3) a correlation will be explored between the number of head impacts (quantified with a notational analysis) and the aforementioned indicators. The study has been registered as a clinical trial (clinicaltrials.gov NCT04841876 (assessed on 16 March 2022)) and has been approved by the Research Ethics Committee of Hong Kong Baptist University (REC/20-21/0452).

All data collection, including motor control, biomechanics, and blood analyses, will be performed through formal collaboration with the universities constituting the Sport and Health Interdisciplinary Group in Movement & Performance from Acute & Chronic Head Trauma (IMPACT) research group based at Cardiff Metropolitan University. Funding for this project will be covered by the institutes involved.

## 2. Materials and Methods

### 2.1. Study Design and Participant Recruitment

This study will be a randomized control trial. The study will consist of two groups: one sport-event group (SG) and one control group (CG); see Figure 1.

Sample sizes were calculated using G*Power 3.1 software (Heinrich-Heine-University Dusseldorf, Dusseldorf, Germany). Calculations were based on a previous study [7], and we aimed to achieve a substantial effect size of 0.8 (Cohen’s d) on blood biomarkers. With an alpha of 0.05 and a power of 80%, 40 participants (20 participants per group) are required to achieve a robust estimation of the relevant parameters. All participants will be required to be over 18 years of age to participate in the study. Prior to data collection, all participants will be fully familiarized with testing procedures and the laboratory environment. In addition, a pilot study will be performed prior to actual experimental data collection. This will provide valuable information concerning timing, standardization, subject preparation, data collection information, logistics, and experimental organization in terms of data management for all participants pre- and post-testing. Members of the research team supervising each different aspect of data collection will ensure the standardization and integrity of data collection. Blood samples will be collected at several phlebotomy locations to ensure bloodletting is performed immediately post-match. There will also be several locations provided by the research team to collect biomechanical and cognitive-motor task data. These multiple locations will enable efficiency and effectiveness in data collection post-event, minimizing the effect of time as a confounding variable.

All the recruited participants will be male rugby 7 s players based at Cardiff Metropolitan University. Inclusion criteria will be participation in rugby 7 s tournaments for a minimum of 2 years and free from injury. After agreeing to participate in this study and passing the eligibility assessment, the participants will sign informed consent forms outlining the experimental design and procedures and will then be randomly assigned to SG or CG in a ratio of 1:1. The control group will comprise 7 s rugby players who are part of the 7 s squad but will not be participating in the actual rugby competition. Therefore, they will be injury- and fatigue-free, and direct comparisons will be possible to investigate the effect of games and contact on sports performance. In addition, baseline measures will also be useful in determining any changes within and between groups. This will enable meaningful comparisons between groups that are sport-specific. Both groups will participate in baseline and post-match data collection. Randomization will be conducted using the random function available in Excel software. All participants will be asked to maintain their regular training and dietary habits during the testing period. The participants will also be required to record their exercises and food intake for 24 h prior to each blood sampling. 

Baseline testing will include measures of (1) demographics, (2) blood, and (3) biomechanical and cognitive-motor tasks and will be conducted on the day before the tournament. The information on the tournament play of each participant in SC, i.e., approximate playing time, number of matches played, and time that the last match was finished, will be recorded and used as the time markers of post-match testing. Post-match testing, including blood and biomechanical and cognitive-motor tasks, will be performed immediately after each game and three times after the third game at 24, 48, and 72 h (Figure 1, Table 1). All data will be collected at the Human Performance Laboratory at Cardiff Metropolitan University. The same order of testing will be standardized for all players based on pilot study results. All experimental data collection will take place immediately post-game for all three matches. Players will report to the laboratory located pitch side for all blood, biomechanical, and cognitive data collection. Once data collection has finished, players will be told to shower and relax. The players will then be asked to return to the laboratory at 24, 48, and 72 h after the tournaments for repeated blood measurements and biomechanical and cognitive-motor tasks. There will be no physical activity during this post-activity testing period.

### 2.2. Measures and Experimental Procedure

#### 2.2.1. Demographics

Age, gender, training duration, medical history, the use of medication, (e.g., mental health issues), health habits (e.g., activity prior to blood draw, alcohol consumption, smoking), rugby playing history, position, and history of concussion will all be recorded at the outset. Personalized anthropometric measures, as well as stature and total body mass, will be recorded for each player using digitized coordinate data from whole-body static pictures. Body segment inertia properties, such as segment masses, moment of inertia, and total body mass center position, will be provided by these measures; the methods described in Gittoes et al., 2009, and Yeadon, 1990 [70,71], will be employed.

The measures and experimental procedures for (2) blood and (3) biomechanical and cognitive-motor tasks and (4) notational analysis are presented below in Section 2.2.2 and Section 2.2.3. The analysis of the data will be combined to address the specific multivariate nature of the research questions.

#### 2.2.2. Blood Methodology

Prior to the competition (the day before), 10 mL of blood will be collected from each player into two separate 5 mL serum separation vacutainer tubes (SSTs) (Becton Dickinson, Rutherford, NJ, USA). These samples will be collected at the same time of day to avoid diurnal effects. Consistency in timing, data collection, and standardization of experimental testing will also provide experimental rigidity and will be established using results from a pilot study prior to data collection.

On competition day, a 10 mL blood sample will be collected into two separate 5 mL SSTs prior to the first rugby game and then again immediately after each subsequent match. Following the final rugby match, blood will be immediately collected, in addition to further sampling at 24 h, 48 h, and 72 h later, also using two separate 5 mL SSTs. All blood samples will be taken using the venipuncture method. Blood will be collected and left to coagulate at room temperature for one hour. After incubation, the blood will be centrifuged at 1250× *g* for 10 min at 4 °C. On completion of centrifugation, the serum supernatant will be collected and kept at −80 °C in two tubes (Eppendorf^®^). Two separate Eppendorf tubes are a requirement, because the serum from both 5 mL SSTs will be kept at Cardiff Metropolitan University for analysis. One tube will facilitate the analysis of serum creatine kinase (CK), cardiac troponin T, and cortisol, while the serum from the other 5 mL SST will be used for the analysis of serum neuron-specific enolase (NSE), protein S100B (S100B), brain-derived neurotrophic factor (BDNF), glial fibrillary acidic protein (GFAP), and ubiquitin carboxyl-terminal esterase L1 (UCHL1). Keeping two separate vials will prevent freeze-thaw cycles and therefore maintain the integrity of the serum samples.

Serum NSE, cardiac troponin T, S100-B, and cortisol will all be measured by an electrochemiluminescence immunoassay using a Modular Analytics E analyzer, supplied by Roche Diagnostics; serum CK will be measured using a Specord 200 spectrophotometer with a Roche/Hitachi 917/MODULAR P analyzer; and BDNF, GFAP, and UCHL1 will be analyzed using an enzyme-linked immunosorbent assay (ELISA).

#### 2.2.3. Biomechanical and Cognitive-Motor Tasks

The methods below outline the procedures for the assessment of biomechanical and cognitive-motor task functions to examine the influence of concussion events during the rugby 7 s tournament. Prior to the rugby tournament, players will complete baseline tests for biomechanical and cognitive-motor tasks. Tests will be performed in a single battery, where players move between stations in a pseudo-random order.

Two categories of tests are explained: (1) whole-body movement and coordination testing and (2) cognitive-motor tasks. The NIRSIT (OBELAB Inc., Seoul, Republic of Korea) Brain Imaging System will be utilized simultaneously with the body movement assessment and cognitive tasks in monitoring, in real time, the brain condition. The NIRSIT is a wireless portable Functional Near-Infrared Spectroscopy (fNIRS) system, which (a) utilizes light to detect hemodynamic changes in the cerebral blood flow and (b) visualizes brain activation regions in the prefrontal area of the brain.

### 2.3. Experimental Procedure

#### 2.3.1. Whole-Body Movement and Coordination Testing

Four whole-body tasks will be completed within the motion capture area: an ecologically valid movement task (i. throwing task), (ii. tackle task), balance (iii. mBESS stance), dynamic balance (iv. tandem gait), and coordination (v. finger-to-nose task).

Throwing task: During the validated movement task, participants will undertake expected and unexpected passing practices in which they will run and pass at top speed to a target (Figure 2). During the anticipated pass, the player will start at a standstill and then will run forward 5 m at full pace to a performing zone, where they will throw a rugby-style pass to the left and right to a target 6 m away; this will be performed 3 times (3-left; 3-right). In the unanticipated task, a light (SMARTSPEED pro, Fusion sport, Australia) in front of the participant will indicate the direction of the pass in a pseudo-random order when they reach 0.5 m from the target line. A successful pass will be one that hits the target, as shown in Figure 2 and as evaluated by a National Performance Director. Data collection: during the whole-body movement and coordination tasks, participants will be labeled using a modified full-body 6-DOF kinematic model [72,73]; using double-sided tape, 38 retroreflective markers (14 mm in diameter) will be applied to the skin (Figure 3). Three-dimensional kinematic (14 cameras, Vicon Vantage Oxford Metrics, Oxford, UK, 250 Hz) and kinetic data will be obtained (force plates: four 600 mm × 900 mm Kistler 9287BA, Winterhur, Switzerland). All cameras will be calibrated to residual errors of <0.3 mm using a 240 mm calibration wand. Prior to further analysis, the external force signals will be amplified internally and aggregated with kinematic data using Vicon Nexus (v2.2.3) and filtered using a low-pass method (4th-order Butterworth, 60 Hz cut-off). Body segment inertia parameters will be based on work by De Leva (1996), as successfully used across a number of dynamic sporting movements [72,73,74].

Tackle task: A tackling drill, where a player performs a maximal tackle on a pad in full-vision and “late vision” conditions with right-left/high-low decision options, will be captured with high-speed video and used to estimate whole-body reaction time, decision making, and aspects of the tackle technique. The late-vision condition provides a constraint on the performer that is more ecologically valid and increases face validity.

mBESS: During the mBESS test, players will be told to stand on their non-dominant foot with their dominant leg kept in 30° hip flexion, 45° knee flexion, and a tandem stance for 30 s with their eyes closed and hands placed on the iliac crest. Players will wear shoes. The non-dominant leg will be determined for all subjects as the non-preferential leg when kicking a ball. Players will be instructed to stand on a force plate located within the capture area.

On completing the mBESS, participants will perform the finger-nose task in the tandem stance position (differs from SCAT 5) with a modified set of instructions from the SCAT5:

“I’m going to put your coordination to the test right now. Please stand on the force plate open your eyes and extend your arm in front of you (shoulder flexed to 90 degrees, elbow and fingers extended). When I provide the start signal, I want you to perform five successive finger-to-nose repetitions using your index finger contacting the tip of your nose, then swiftly and accurately returning to the starting position”.

Tandem gait: Players will be told to stand behind a starting line with their feet together and their shoes on. Players will walk heel-to-toe, with one foot’s toes contacting the heel of the next at each step, for 3 m (eyes open), and then turn 180° and return (total 6 m). A 38 mm wide (sports tape) 3 m line will outline the route on the floor. Participants will be corrected and asked to continue if they step off the line, have a space between their heel and toe, or touch or grab the examiner or an object.

Finger-to-nose: A finger-to-nose task will be performed within the data-capturing area in accordance with a modified SCAT 5 procedure, with the subject standing with both feet on the force plate. “I’m going to put your coordination to the test right now. Stand on both feet, eyes open, and your dominant arm extended in front of you. Perform five sequential finger-to-nose repetitions, touching the tip of the nose with your index finger on the start signal, then returning to the starting position. Execute the actions as quickly and properly as possible”. Cognitive-motor tasks are covered in Section 2.3.2.

Prior to the cognitive-motor exam, participants will take the Mini-Mental State Examination (MMSE), which consists of 30 questions that assess participants’ sense of orientation, attention, immediate and delayed recollection, language, and construction [75,76].

In order to examine cognitive function, a series of tasks will be performed (i. Pegboard; ii. Tapping Task; iii. Stroop word test; iv. reaction time task; v. working memory).

Purdue Pegboard: The Purdue Pegboard is used to explore bimanual finger and hand dexterity [66,67]. The Purdue Pegboard consists of two parallel rows of 25 holes, with pegs first inserted into two laterally positioned cups, followed by collars and washers first pushed into two central cups. Four subtests are performed. The first subtest entails placing as many pins as possible in the holes (from top to bottom) within a 30 s time constraint, first with the favored hand, then with the non-preferred hand, and finally with both hands. In the fourth subtest, the individual alternates between using both hands to construct “assemblies” of a peg, a washer, a collar, and another washer in 1 min.

Tapping Task: The finger-tapping test (FTT) is a lateralized coordination and motor speed test. To conduct the test, the palm will be taped to a flat board with fingers splayed. The index finger will then “tap” on a computer keyboard button. Participants will be instructed to tap each finger on the button as quickly as possible throughout a 10 s period. There will be three trials with each finger.

To increase the number on the counting device with each tap, the number of taps will be averaged over trials [77].

Stroop word test: The Stroop color test will be carried out on a computer with the help of the online demo test [78]. Participants will be seated comfortably at the computer workstation and will be instructed on how to conduct the test. The test will be a demonstration and take 2 min to complete. Participants will be required to say the words aloud, since verbal and cognitive performance has been reportedly related to mild concussion symptoms. Stroop will be run using both vocal and keyboard responses. The four colors red, blue, green, and yellow will be utilized to convey 40 words for 2 s each. To allow scoring after the test, a screen + camera capture of the Stroop word and the participant response will be used for the spoken response. The view of the video will include the participant and the Stroop word on the screen. In the keyboard response task, participants will respond to the four colors using the r, b, g, and y keys on the keyboard. We will then explore whether there is a correlation between keyboard and verbal scores and if one of these modes of answering is more affected by a blow to the head.

Reaction time task: To quantify reaction time, a computerized reaction time test will be performed. The test will be repeated 5 times with each hand; more information on the reaction time challenge may be found here [79].

Working memory: In the Digit Span Backward test, participants are shown a string of digits that must be repeated backward. According to the SCAT 5, the working memory score is determined from the temporal reactions to rising number strings [53]. The raw data will be used to calculate working memory.

#### 2.3.2. Sports Performance Analysis

Sports performance analysis of training games will occur via computerized observation using a HUDL Elite software package (Version 11, SportsCode, Warriewood, NSW, Australia). This software will be used to create a workflow analyzing tactical and physical elements of performance, including collisions, impacts, and bouts of high-speed running (HSR). Collision measurement will occur after sessions and will be undertaken by two experienced performance analysts. The collision count will be deemed to include the sum count of all ball carries into contact, rucks, mauls, and, most importantly, tackles. All sessions will be recorded using a digital camera system.

Evidence-based preparation in elite rugby union has led to the increasing use of athlete tracking. External loads generated by the players will be calculated using accelerometers, local positioning systems (LPSs), global positioning systems (GPSs), and optical tracking systems [80]. Body load and impact measurements can be used to quantify the amount and intensity of any physical encounters and collisions between athletes, objects, or surfaces. Any serious injuries to the players during the three-game period will be managed immediately by the medical team.

Each player will wear a GPS micro-technology unit (mass = 67 g, 10 Hz V5.0 and 10 Hz S5, Catapult Innovations, Scoresby, VIC, Australia) in a pocket fitted in his/her playing jersey on the upper thoracic spine between the scapulae. Each GPS unit collects data at a rate of 10 hertz. The dependability of the device has already been proven to be valid and reliable for measuring speed and distance in team sports [81].

Because of their training and competitive routines, all players are completely familiar with the units. Throughout the data-collecting period, each player will wear the same assigned GPS unit. To ensure a comprehensive and high-quality satellite signal, GPS units will be turned on at least 10 min before the game. After the game, GPS data will be downloaded to a laptop and analyzed using Sprint 5.1 software (Catapult Innovations, Scoresby, VIC, Australia). Body load (measured in G-force) is the sum of all forces applied to an athlete, including acceleration/deceleration, related changes in direction, and impacts from both player-to-player and ground contact (foot strikes and falls) (Table 2).

### 2.4. Data Processing

#### 2.4.1. Whole-Body Movement and Coordination Testing

Trajectories will be examined using Visual 3D (v6, C-Motion Inc., Germantown, TN, USA) after markers have been labeled. Using residual analysis, raw marker coordinates will be low-pass-filtered (4th-order Butterworth) with cut-off frequencies of 12 Hz and 8 Hz. Joint angle data for flexion-extension, internal-external rotation, and abduction-adduction (x-axis, z-axis, and y-axis, respectively) will be determined as the transformation between two segment coordinate systems (SCSs) given by an X-Y-Z Cardan sequence of rotations. Extension/plantarflexion and flexion/dorsiflexion will be represented by positive and negative angles, respectively. A variant of Ridder’s split difference method will be used to differentiate linear and angular values [83]. Newton-Euler inverse dynamics algorithms will be used to calculate the generated joint moments. The product of joint power will then be determined.

Movement task (i.e., throwing): Joint angular kinematics, kinetics, coordination, and coordination variability will be assessed. Movement coordination will be captured using vector coding and principal component analysis. Characteristics of a candidate collective variable for throwing will also be included in the analysis, described by the phase relation between the total body mass center of wrist location. The analysis of balance (mBESS stance), dynamic balance (tandem gait), and coordination (finger-to-nose task) are based on published clinical measures.

#### 2.4.2. Cognitive-Motor Tasks

The cognitive-motor tasks will be examined using the standard clinical procedure. Specifically, Stroop word test: Scores will be reported as the average response time in incompatible trials minus compatible trials. Reaction time task: The task will be scored digitally in milliseconds. Purdue Pegboard: The number of pins inserted in the time period for each hand determines the score for the pin (peg) insertion subtests. The total number of pairs of pins inserted determines the score for the bimanual condition. The assembly score is the number of components assembled (for more information on how to set up/run a test, check the extended usage instructions in the Scoring Application or on our website). Finally, Tapping Task: Timing, consistency, and frequency will be quantified.

### 2.5. Statistical Analysis

Data will be analyzed using IBM SPSS 26.0. Diagnostic testing (e.g., outlier detection, distribution examination, homogeneity checking) will be conducted for all data, and all skewed values and missing values will be addressed using a multiple imputation approach. To examine the differences between the baseline and the testing days for each of the physiological, cognitive-motor, and whole-body movement skills, a series of generalized linear mixed models will be employed, controlling for several demographic covariates. Post hoc tests will be performed using the Bonferroni procedure, Mann-Whitney U test, and Chi-square test for different types of data. The significance level will be set at 95% (two-tailed). The effect size of Cohen’s d will be calculated, with 0.3, 0.5, and 0.8 indicating small, medium, and large effect sizes, respectively.

Due to the difference in the type of data (discrete and continuous), a variety of methods will be employed, including statistical parameter mapping to examine continuous data between the four sessions and baseline. A canonical analysis will be used as a multivariate technique to determine the relationships between the variables and the selected performance of key dependent variables. All collected data will be anonymized and stored on a password-protected computer, with access only granted to the lead investigator.

## 3. Discussion

Sports-related concussion (SRC) diagnosis is hindered by a lack of validated, objective diagnostic tests. There is an urgent requirement to understand and examine sports-related head trauma and the possible short-, medium-, and long-term consequences of a concussion. Using an interdisciplinary approach, this research will examine the changes in the blood-based biomarkers and biomechanical and cognitive-motor abilities of elite rugby players across a three-match tournament. The IMPACT Interdisciplinary Group’s purpose is to examine a number of functional biomechanical and motor control tests and biochemistry to investigate changes resulting from contact sports.

We expect a decline in coordination, cognitive-motor tasks, and whole-body mobility, owing to exhaustion and/or concussions, as the number of games increases. Concussive events and injuries will be assessed using a notational analysis recorded during each match. Specifically, a reduction in the performance variable associated with the mBESS stance and finger-to-nose task is expected. In the analysis of the emergent coordinative structures of the passing task, we expect to show a shift to a critical level, before which the outcome variable of passing accuracy will be maintained. Past this critical point, the self-organization of the biological structures will be unable to remain stable, and the coordination variability will rise above the functional level. Examining the structure and timing of these movement characteristics will allow us to link the dynamics to the evolving rugby games. Cognitive function tests, including the Pegboard, Tapping Task, Stroop word test, and reaction time task, are expected to display a reduction in outcomes in the repeated testing sessions as a result of fatigue and/or a concussive event. With respect to the blood-based biomarkers, we expect CK, cardiac troponin T, NSE, S100B, BDNF, GFAP, and UCHL1 to increase after a rugby match, as exercise and contact sports have been shown to cause these biomarkers to increase [8,9,10,11,12,13,14,15,16,17,18,19,20,21,22,23,24,25,26,27,28,35,37,44,45,46,47,48,49]. However, cortisol concentrations may not increase, as cortisol has been shown not to increase after American football play in the absence of brain injury [10,37]. Some variations in biomarker concentrations in the blood are expected to correspond to changes in biomechanical and cognitive-motor activities. In the examination of the interaction of whole-body movement, coordination testing, cognitive-motor tasks, and blood-based biomarkers, we expect to observe those variables most related to changes in the testing session and the other outcome variables, including the accuracy of passing.

The study proposed here will produce significant, validated research findings on the frequency of head trauma events and their impact on the performer’s biological system, with clear implications for doctors, scientists, and sports-regulating authorities. The implications of these findings will provide valid information that may result not only in a safer rugby environment but also in a shift in athleticism and a reconsideration of the rules of the game, with fewer impacts, collisions, and contact phases. Further implications may include the consideration of a technical fitness assessment after a concussive event. Players may be medically fit to return to play following injury, but skill levels may have deteriorated due to cognitive decline, resulting in the deterioration of cognitive abilities and skills. This suggestion may explain in part the reoccurrence of head injury linking concussive events to future concussive events. Therefore, medical examinations and comprehensive cognitive ability assessments may be required prior to returning to play. This study’s scope will be significant, and its consequences for health and well-being in athletic communities of all levels, genders, and ages will offer researchers a better knowledge of human responses to trauma. Through the dissemination of knowledge by publishing papers in international peer-reviewed journals, presenting at international conferences, and sharing our findings with the medical profession, governing bodies of sports, and coaches and trainers, the IMPACT group will provide meaningful information to drive the next generation of researchers investigating concussion in sports. Embracing an interdisciplinary approach allows a holistic view and novel conceptual understanding of this paradigm.

## Figures and Tables

**Figure 1 biology-12-01089-f001:**
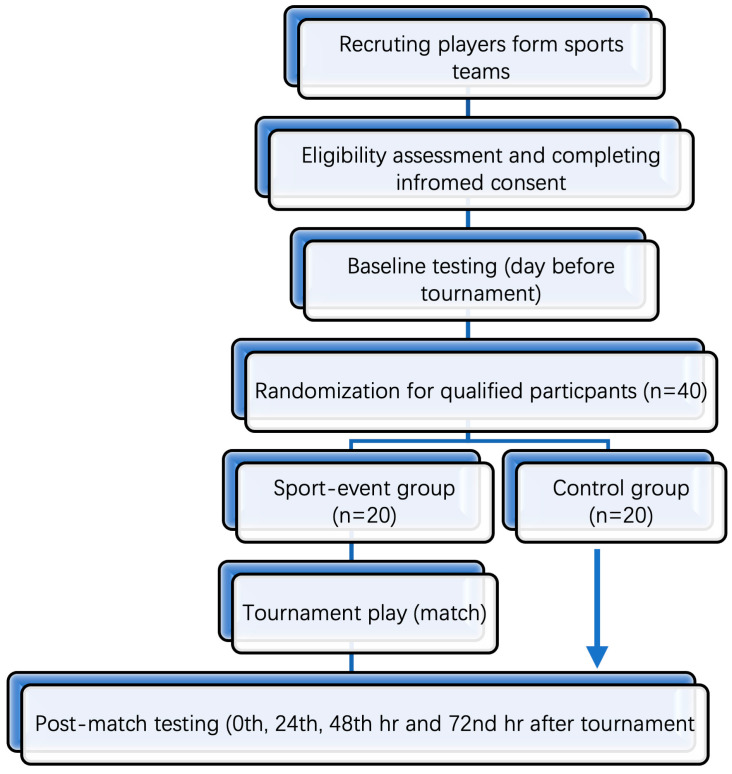
Flow diagram outlining timeline of events used in the testing protocol.

**Figure 2 biology-12-01089-f002:**
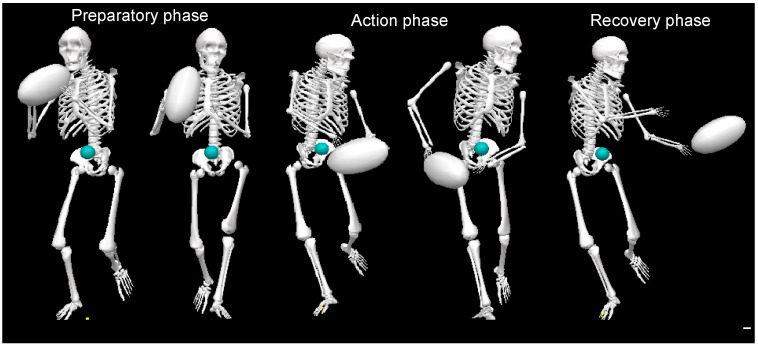
Rugby pass performed during the whole-body movement and coordination testing showing the preparatory, action, and recovery phases.

**Figure 3 biology-12-01089-f003:**
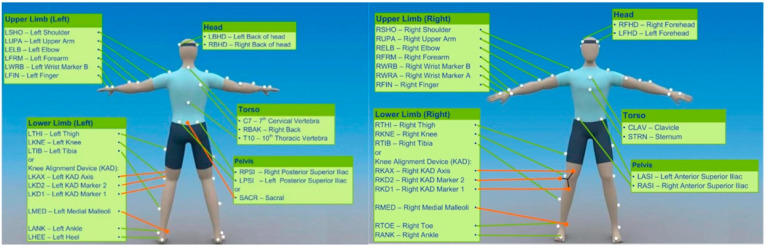
Marker locations of the 6-degree-of-freedom kinematic model.

**Table 1 biology-12-01089-t001:** Timeline for testing showing the baseline, before game, and three post-game assessments.

Base Line		Testing Day	
Day before Testing Day	Post-Game 1	Post-Game 2	Post-Game 3
**Demographic information:**			
Standing height, age, sex, total body mass, duration of training, medication use, medical history			
**Anthropometric Characteristics**			
Body segment inertia characteristics			
**Bloods:**	**Bloods:**	**Bloods:**	**Bloods:**
Serum creatine kinase, cardiac troponin, cortisol; neuron-specific enolase, protein S100B, and brain-derived neurotrophic factor.	Serum creatine kinase, cardiac troponin, cortisol; neuron-specific enolase, protein S100B, and brain-derived neurotrophic factor.	Serum creatine kinase, cardiac troponin, cortisol; neuron-specific enolase, protein S100B, and brain-derived neurotrophic factor.	Serum creatine kinase, cardiac troponin, cortisol; neuron-specific enolase, protein S100B, and brain-derived neurotrophic factor.
**Biomechanics**	**Biomechanics**	**Biomechanics**	**Biomechanics**
**Whole body movement and coordination testing**	**Whole body movement and coordination testing**	**Whole body movement and coordination testing**	**Whole body movement and coordination testing**
An ecologically valid movement task (i. Throwing ask), Balance (ii. mBESS stance), Dynamics balance (iii. tandem gait) and coordination (iv. Finger to nose task).	An ecologically valid movement task (i. Throwing ask), Balance (ii. mBESS stance), Dynamics balance (iii. tandem gait) and coordination (iv. Finger to nose task).	An ecologically valid movement task (i. Throwing ask), Balance (ii. mBESS stance), Dynamics balance (iii. tandem gait) and coordination (iv. Finger to nose task).	An ecologically valid movement task (i. Throwing ask), Balance (ii. mBESS stance), Dynamics balance (iii. tandem gait) and coordination (iv. Finger to nose task). Collected at 0, 24, 48 and 72 h post Game 3
**Cognitive-motor tasks**	**Cognitive-motor tasks**	**Cognitive-motor tasks**	**Cognitive-motor tasks**
Cognitive functioning test using the Mini Mental State Examination (MMSE)	Cognitive functioning test using the Mini Mental State Examination (MMSE)	Cognitive functioning test using the Mini Mental State Examination (MMSE)	Cognitive functioning test using the Mini Mental State Examination (MMSE). Collected at 0, 24, 48 and 72 h post Game 3
i. Pegboard, ii. Tapping Task, iii. Stroop word test, iv. Reaction time task, v. Working memory	i. Pegboard, ii. Tapping Task, iii. Stroop word test, iv. Reaction time task, v. Working memory	i. Pegboard, ii. Tapping Task, iii. Stroop word test, iv. Reaction time task, v. Working memory	i. Pegboard, ii. Tapping Task, iii. Stroop word test, iv. Reaction time task, v. Working memory. Collected at 0, 24, 48 and 72 h post Game 3
	**Notational Analysis**	**Notational Analysis**	**Notational Analysis**
	Performance related analysis: to include X, Y, Z	Performance related analysis: to include X, Y, Z	Performance related analysis: to include X, Y, Z

**Table 2 biology-12-01089-t002:** Adapted from Gabbett (2013) [82], zones used to study “impacts” during rugby games 1–3.

Zone	Gravitational Force	Description of Impact
**1**	<5.0–6.0	Very light impact, hard acceleration/deceleration/change of direction while running
**2**	6.1–6.5	Light to moderate impact, minor collision with opposition player, contact with ground
**3**	6.5–7.0	Moderate to heavy impact, making tackle or being tackled at moderate velocity
**4**	7.1–8.0	Heavy impact, high-intensity collision with opposition player/s, making direct front on tackle on opponent traveling at moderate velocity, being tackled by multiple opposition players when running at submaximal velocity
**5**	8.1–10.0	Very heavy impact, high-intensity collision with opposition player/s, making direct front on tackle on opponent traveling at moderate velocity, being tackled by multiple opposition players when running at near maximal velocity
**6**	>10.1	Severe impact, high-intensity collision with opposition player/s, making direct front on tackle on opponent traveling at moderate velocity, being tackled by multiple opposition players when running at maximal velocity

## Data Availability

Not applicable.

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
