# Peer review of "Sports-Related Concussion Assessment: A New Physiological, Biomechanical, and Cognitive Methodology Incorporating a Randomized Controlled Trial Study Protocol"

_biology, 2023, doi:10.3390/biology12081089_

Round 1

Reviewer 1 Report

The proposed protocol is very interesting and should be very useful for consussion assessment. However several issues should be rewriten in order to improve the protocol understanding.

The simple summary appears to describe a different aim for this paper.

In introduction in paragraph in lines 172 to 184 talks about the relationship of the number of impacts but perhaps the relationship with their intensity (measured with GPS systems) should be considered and included.

In methodology section Control Group activity is not described. It had been interesting to defined three groups (sport without contact group, sport with contacts group and control group) in order to analyze the different markers and biomechanical variables changes due to the exercise or due to the impacts. 

The same protocol of measurements will be applied to control group? Perhaps, it does not make sense to subject who does not do any sport activity (CG)  to suffer from exccessive blod extractions and test, not waiting any change in the parameters. It would not be ethical.

Going into more detail in the measurements, in line 242 a "digital coordinate data" is not described in detaid with no information about the application of this measurement data.

Paragraph in lines 245-248 does not provide information.

The methodology section would be rewriten in order to organize the information. It would be better to describe the methodology by type of tests all together including the processin of each test next to the procedure. The order of the test description  could be in order of the assessment protocol.

The variables of the study could be summarized in a table according to the type of variable (kinematic, cognitive...).

The are several test that are not described with enougth details. The use of NIRSIT it is not explained (which variables will be used? How the use of this system will be implemented in the biomechanical and cognitive tasks? 

In line 300 the SMART SPEED system is named by not explained (neither referenced). 

The tackle task (lines 320 to 323) is not well described, what is "late vision" condition? Not references has been added to explain better this test.

Paragraphs from 328 to 351 will be rewriten avoiding to include  the test of SCAT 5. Better to indicate that the oral instructions of the test have simply been modified from the original ones.

More information about the Reaction time task should be added. 

In statistical analysis a better description of the will improve the understanding of the analysis proposed.

In discussion, references to other studies about the effect of the contact in the different variables analyzed should be included. 

Author Response

The proposed protocol is very interesting and should be very useful for consussion assessment. However several issues should be rewriten in order to improve the protocol understanding.

The simple summary appears to describe a different aim for this paper.     

Thank you for your comment we have added a context sentence has been added to the simple summary to ensure parity of the aim.

In introduction in paragraph in lines 172 to 184 talks about the relationship of the number of impacts but perhaps the relationship with their intensity (measured with GPS systems) should be considered and included.

Thank you for your comment we have added intensity to this sentence

In methodology section Control Group activity is not described. It had been interesting to defined three groups (sport without contact group, sport with contacts group and control group) in order to analyze the different markers and biomechanical variables changes due to the exercise or due to the impacts. 

Thank you for you comment a section has been added in the methodology

The same protocol of measurements will be applied to control group? Perhaps, it does not make sense to subject who does not do any sport activity (CG)  to suffer from exccessive blod extractions and test, not waiting any change in the parameters. It would not be ethical.

Thank you for you comment a section has been included

Going into more detail in the measurements, in line 242 a "digital coordinate data" is not described in detaid with no information about the application of this measurement data.

Thank you for your comment, more detail has been added

Paragraph in lines 245-248 does not provide information.

Thank you for your comment, information has been added for clarification. 

The methodology section would be rewriten in order to organize the information. It would be better to describe the methodology by type of tests all together including the processin of each test next to the procedure. The order of the test description  could be in order of the assessment protocol.

Thank you for your comment, however we feel the methods do reflect the organisation of this protocol.

The variables of the study could be summarized in a table according to the type of variable (kinematic, cognitive...).

Thank you for your comment. Because this is a new protocol, we feel that the format is adequate in relation to explanation of the variables.

The are several test that are not described with enougth details. The use of NIRSIT it is not explained (which variables will be used? How the use of this system will be implemented in the biomechanical and cognitive tasks? 

Thank you for your comment, the variables are outlined in the methods section.

In line 300 the SMART SPEED system is named by not explained (neither referenced). 

Thank you for your comment, SMART speed has been referenced (Smartspeed pro, Fusion sport, Australia), it is a well known industry standard brand. 

The tackle task (lines 320 to 323) is not well described, what is "late vision" condition? Not references has been added to explain better this test.

In addition, ‘late vision’ refers to the tackle pad being introduced later in the skill, hence constraining of the player. This has been added to the methods

Paragraphs from 328 to 351 will be rewriten avoiding to include  the test of SCAT 5. Better to indicate that the oral instructions of the test have simply been modified from the original ones.

Thank you for your comment, in order to provide enough detail regarding the modified SCAT 5 we have left the verbal instructions, we believe this will help in the replication of the methodology.

modified SCAT 5 protocol

More information about the Reaction time task should be added. 

Thank you for your comment, we have provided a clearer link to the reaction time task reference. 

In statistical analysis a better description of the will improve the understanding of the analysis proposed.

Thank you for your comment, we have checked the statistics section to ensure this is an accurate and clear section

In discussion, references to other studies about the effect of the contact in the different variables analyzed should be included. 

Because this is a new protocol, we feel that this section should be discussed following experimental data collection.

Reviewer 2 Report

This study protocol aims to explore the novel sports-related concussion assessment. This study protocol is quite intriguing and important for sports scientists, coaches, and clinicians working in sports. It was well-developed and elaborated. Some specific comments can be found below.

Abstract:

1.     I suggest adding the implications for sports coaches and managers since they are making the final decision about returning to the play.

Introduction:

1. The Introduction is rather well-written, with appropriate rationale and literature review. However, I suggest that authors improve the linkage between paragraphs to further improve the readability of the introductions. Now it looks like the paragraphs are simply tossed one after another.

Methods:

1. Did the authors consider the potential dropout of the participants? Since G*Power showed that 15 participants per group are needed, a potential dropout that usually happens in this research might compromise this. I believe more participants need to be recruited in the initial phase to end up with 15 participants per group.

2. I`m concerned that some of the proposed tests are not specific enough and that a more rugby-specific test can be used. For example, numerous studies showed low sensitivity in reaction time between elite and non-athletes if the task or stimulus needs to be more specific. Therefore, a more specific reaction time test might be more sensitive to detect the potential effects of concussions. For example, start and acceleration (running) after a sound, visual stimulus, or similar specific test might be more ecologically valid. See some ideas here: https://www.tandfonline.com/doi/abs/10.1080/24748668.2015.11868852

3. Please provide a reference for applying multiple imputations in sport science and medical research.

4. All in all, methods are written in a precise and highly scientific manner.

Discussion:

1. The discussion is brief and adequate for this kind of study.

Author Response

Abstract:

  1. I suggest adding the implications for sports coaches and managers since they are making the final decision about returning to the play. 

Thank you for your comment, we have added coaches and managers to the implications sentence.

Introduction:

  1. The Introduction is rather well-written, with appropriate rationale and literature review. However, I suggest that authors improve the linkage between paragraphs to further improve the readability of the introductions. Now it looks like the paragraphs are simply tossed one after another.

Thank you for your comment, we have ensured that introduction more clearly flows

Methods:

1. Did the authors consider the potential dropout of the participants? Since G*Power showed that 15 participants per group are needed, a potential dropout that usually happens in this research might compromise this. I believe more participants need to be recruited in the initial phase to end up with 15 participants per group. 

Thank you for your comment, we agree and have include a buffer for the potential drop out.

  1. I`m concerned that some of the proposed tests are not specific enough and that a more rugby-specific test can be used. For example, numerous studies showed low sensitivity in reaction time between elite and non-athletes if the task or stimulus needs to be more specific. Therefore, a more specific reaction time test might be more sensitive to detect the potential effects of concussions. For example, start and acceleration (running) after a sound, visual stimulus, or similar specific test might be more ecologically valid. See some ideas here: https://www.tandfonline.com/doi/abs/10.1080/24748668.2015.11868852

Thank you for your comment, we are using currently available tests for this study that are valid and reliable. While we agree that there should be more rugby orientated specific tests, the design of such tests would require a separate study. In addition, the tests used here can provide comparisons in the literature

  1. Please provide a reference for applying multiple imputations in sport science and medical research.

Thank you for your comment, This is not available at the moment as the protocol is new and this methodology has not been used previously.

  1. All in all, methods are written in a precise and highly scientific manner.

Thank you for your comment.

Discussion:

  1. The discussion is brief and adequate for this kind of study.

Thank you for your comment

Reviewer 3 Report

General Comments

This is an interesting manuscript, presenting the design of a randomized control trial study incorporating physiological, biomechanical, cognitive and sport performance analysis methods for evaluating the rugby players after the tournaments. The Authors planned to compare a traumatic sport-event group (n=15) and a control group (n=15) of male rugby players (aged > 18 years) before (testing baseline) and immediately after each three games, and at 24, 48, and 72 hours after the third (last) game. All planned measures of demographics, blood analysis, biomechanical and cognitive-motor tasks, and sport performance analysis are correctly presented and analyzed in Discussion. The Authors expected that the changes in blood-based biomarker concentrations will  correlate with changes in biomechanical and cognitive-motor tasks in rugby players They also prognoses that proposed study will yield meaningful, ecologically valid data regarding the occurrence of concussive events during game demands, and the influence of these events on the biological systems of the performers. This will facilitate a greater understanding about the human responses to exercise induced trauma.

The manuscript is generally well written. However, the design of this paper should be little improved before publishing.

1.The authors should be: (1) To rewrite the aim of this study (in Abstract and at the end of Introduction), changing future form to past form, and clearly indicating that this is design (the research project without results) of a randomized control trial study incorporating physiological, biomechanical, cognitive and sport performance analysis methods for evaluating the rugby players after the tournaments (or something similar). In the present form, the aim of this study does not characterize adequately the presented topic of this manuscript (research project). (2) To present Conclusion in the Abstract and at the end Discussion, indicating the main expected (possible) novel effect of this designed research project for measurement of adaptive changes in rugby players with and without a sport-specific trauma after the planned tournaments.

(3) To describe more detail in Discussion how a formal collaboration with the Hong Kong Sports Institutes and the Universities comprising the Sport and Health Interdisciplinary group in Movement & Performance from Acute & Chronic head Trauma (IMPACT) research group is planning to realise in the framework of the present project.  

2. In my opinion, it is obligatory to add as a limiting factor of the study the facts, that body composition (fat distribution across the body, lower extremity (or appendicular) lean mass, muscle mass of the lower extremities, etc.) is not planned to measure in this this study which cause some difficulties to interpret the results of this study. This notice  should be mentioned and analysed at the end of the Discussion.

Specific Comments

Abstract (Page 1)

(1)  Please rewrite the aim of this study (see General Comments).

(2)  Please add Conclusion at the end of Abstract (see General Comments).

1. Introduction

Page 4. Please rewrite the aim of this study (see General Comments).

3. Discussion

Page 12. Please describe more detail the collaboration between the Hong Kong Sports Institutes and the Universities comprising the Sport and Health Interdisciplinary group in Movement & Performance from Acute & Chronic head Trauma (IMPACT) research group in the framework of this designed project.

Page 12. Please describe limitations of this study at the end of Discussion (see General Comments).

Author Response

Thank you very much for your comments.

Round 2

Reviewer 1 Report

Thank you for the modifications made in the paper. I still have several comments that I have not noticed before.

In methods, why do yo measure the same parameters in base line and before game condition? Which is the objetive of these measurements? Due to the fact that the protocol is "so intense ", it should be better not to add an additional measurement day that maybe does not contribute to the study. Moreover the previous game assesment does not appear in Fig. 1.

The number of participants in line 195 and in Fig 1 is different. Please modify figure 1.

As I commented before, the use of a control group that would do an sport without contact would allow to identify the changes in the different variables due to the exercise and not due to the contacts. In other case you will not know  the reason of the changes. 

How do you could solve this?

Author Response

We thank the reviewer 1 and editorial team for their useful help and comments. We have addressed all issues as outlined below and highlighted in the paper in red text.

1. In methods, why do yo measure the same parameters in base line and before game condition? Which is the objetive of these measurements? Due to the fact that the protocol is "so intense ", it should be better not to add an additional measurement day that maybe does not contribute to the study. Moreover the previous game assesment does not appear in Fig. 1.

Thank you for your comments. We have addressed this issue and now will only be taking bloods and measures at baseline and post the three games. This has been modified in the table.

2. The number of participants in line 195 and in Fig 1 is different. Please modify figure 1.

Thank you for your comments. We have changed this in the text and in the figure. There are now 40 subjects 20 in the experimental group and 20 in the control.

3. As I commented before, the use of a control group that would do an sport without contact would allow to identify the changes in the different variables due to the exercise and not due to the contacts. In other case you will not know  the reason of the changes. 

Thank you for your comments. We have included a section on this in the paper highlighted in red. The contact group will be compared to the control group who will not be participating in the games. The control group will also be rugby players. This will provide direct comparisons that are sport specific. In addition, changes in the experimental group can also be compared to their own baseline measures obtained prior to the games.